# Determinants of Antenatal Education and Breastfeeding Uptake in Refugee-Background and Australian-Born Women

**Tam Anh Nguyen** [1], **Mohammed Mohsin** [2,3], **Batool Moussa** [2], **Jane Fisher** [4], **Nawal Nadar** [2], **Fatima Hassoun** [2], **Batoul Khalil** [2], **Mariam Youssef** [2], **Yalini Krishna** [2], **Megan Kalucy** [2] and **Susan Rees** [2,*]

[1]  Faculty of Medicine & Health, University of New South Wales, Sydney, NSW 2052, Australia; tamanhalice@gmail.com
[2]  Discipline of Psychiatry and Mental Health, School of Clinical Medicine, Faculty of Medicine & Health, University of New South Wales, Sydney, NSW 2052, Australia; m.mohsin@unsw.edu.au (M.M.); b.moussa@unsw.edu.au (B.M.); n.nadar@unsw.edu.au (N.N.); f.hassoun@unsw.edu.au (F.H.); b.khalil@unsw.edu.au (B.K.); m.yousif@unsw.edu.au (M.Y.); yalini.krishna@unsw.edu.au (Y.K.); m.kalucy@unsw.edu.au (M.K.)
[3]  Mental Health Research Unit, Liverpool Hospital, NSW Health, Sydney, NSW 2170, Australia
[4]  Global and Women's Health Unit, Public Health and Preventive Medicine, Monash University, Clayton, VIC 3800, Australia; jane.fisher@monash.edu
*  Correspondence: s.j.rees@unsw.edu.au

**Abstract:** Despite the well-established benefits of antenatal education (ANE) and breastfeeding for mothers, there is a paucity of evidence about the uptake of ANE and breastfeeding amongst women from refugee backgrounds or its associations with sociodemographic factors. The current study is a cross-sectional survey at two time points examining the prevalence of ANE attendance, breastfeeding, and intimate partner violence (IPV) amongst 583 women refugees resettled in Australia and a control group of 528 Australian-born women. Multi-logistic regression was used to explore bivariate associations between ANE attendance, breastfeeding, IPV, and sociodemographic characteristics (parity, maternal employment, and education). Refugee-background women compared to Australian-born women have lower ANE utilization (20.4% vs. 24.1%), higher rates of breastfeeding on hospital discharge (89.3% vs. 81.7%), and more IPV reports (43.4% vs. 25.9%). Factors such as nulliparity, higher level of education, and employment predict higher rates of ANE and breastfeeding adoption. In contrast, IPV is a risk factor for ANE underutilization. Further, of the women from refugee backgrounds who accessed ANE services, 70% attended clinics designed for women from non-English-speaking backgrounds. These findings support the need to ensure effective screening and interventions for IPV during antenatal care and to better understand the role of culture as a protective or risk factor for breastfeeding initiation.

**Keywords:** antenatal education; intimate partner violence; breastfeeding; refugee; employment; women

## 1. Introduction

In Australia, antenatal education (ANE) is considered an important component of antenatal care. ANE is offered in group classes to women at most public and many private hospitals [1]. These educational programs typically focus on education to develop knowledge, skills, and confidence in understanding pregnancy, the birth process, and the hospital setting. The broad goal is to prepare the woman, her partner, and her family, if appropriate, for childbirth and early parenting [1]. Women who attend ANE classes are associated with better adjustment to parenthood [2], lower rates of negative birth outcomes, reduced maternal stress, and use of interventions (including cesarean section and epidural anesthesia) [3,4]. Furthermore, it has been identified that to improve health outcomes, culture- or language-specific ANE programs should be offered for women from mainly non-English-speaking or immigrant backgrounds [5]. Multicultural health workers play a vital role in facilitating referrals to these specialized services and promoting community development [6].

Despite better health equity and universal healthcare coverage provision in high-income countries (HICs), such as Medicare in Australia, which provides free antenatal care amongst other health services [6], women who arrive as migrants (both forced and economic) have lower rates of participation in antenatal care, where pregnancy-related screening and monitoring occur and social factors relevant to the pregnancy are discussed [7–11]. Poor, late, or lack of antenatal appointment attendance has been linked to negative birth outcomes in both groups of women [7,10]. While being of refugee background poses a risk factor for both adverse post-partum health outcomes and antenatal services underutilization [7], there exists a scarcity of empirical research on the impact and outcomes of ANE, as distinct from the general antenatal services participation, among refugee-background women in comparison to women born in HICs. In addition to immigration status, data from North America, Australia, and Europe have revealed that low income, low socioeconomic status (SES), and limited educational attainment contribute to lower antenatal services utilization and engender maternal health inequity [12,13]. Given this well-established association between socioeconomic factors and health outcomes, as well as the fact that women from refugee backgrounds often have lower SES compared to their native-born counterparts [14,15], it is imperative to explore the disparities between these two cohorts and factors contributing to their health trajectories.

As part of routine antenatal care, intimate partner violence (IPV) screening is typically conducted as an essential component of risk assessment [6]. In the antenatal setting, IPV often remains inadequately acknowledged, despite its higher prevalence than other obstetric risks such as pre-eclampsia or gestational diabetes [16]. Intimate partner violence is defined as any behavior by a current or former intimate partner that involves physical, sexual, financial, or psychological harm [17,18]. According to the World Health Organization (WHO), one-third of women worldwide have been subjected to IPV in their lifetime [19], with the highest rate observed in regions of Oceania and Sub-Saharan Africa (29–32%) and the lowest reported in central Europe (16%) [20]. IPV during pregnancy increases the risk of miscarriage, stillbirth, pre-term labor, and low-birth-weight neonates [16,21]. The evidence also suggests that IPV perpetrated primarily against the mother has has developmental effects on the child, such as increased risk of insulin resistance, psychiatric disorders, low intellectual capability, and cognitive impairment [22]. Predisposing factors include IPV-related prenatal insults such as stress, substance abuse, inadequate nutritional intake, poor antenatal services utilization, and infective agents. During infancy, witnessing IPV and associated parental stress can have negative effects on socio-emotional development [22–26]. Women from refugee backgrounds experience higher rates of psychological and/or physical IPV during pregnancy compared to women born in the host country, with reported prevalence of 44.4% and 25.8%, respectively [27]. Whilst general perinatal care routinely includes IPV screening in most Australian jurisdictions, refugee-background women are more likely to experience barriers and challenges accessing that care. Furthermore, lack of trust and differences in expectations and communication styles from those of healthcare providers, may result in the underreporting of IPV [27,28].

Promoting early and exclusive breastfeeding is a central feature of ANE. Early and long-term breastfeeding, including exclusive breastfeeding for six months and non-exclusive continuation for two years, is recommended due to its numerous health benefits for both babies and mothers [29]. Early initiation of breastfeeding has been linked to a two-fold reduction in the mortality of infants across countries [30]. Breastfed babies demonstrate greater immunity, with lower odds of infectious morbidities such as gastrointestinal diseases, respiratory infections, otitis media, and urinary tract infections [31]. With respect to the child's long-term outcomes, breastfeeding improves cognitive function and performance on intelligence tests [32], and it protects against type 2 diabetes [29,33]. For mothers, breastfeeding is associated with decreased risk of maternal depression, breast and ovarian cancer [29], endometrial cancer, osteoporosis [34], and strokes amongst postmenopausal women [35]. Additionally, there is strong evidence of positive maternal–infant bonding associated with breastfeeding [34]. The current evidence shows that while breastfeeding

is initiated in 98% of infants soon after birth [36], the duration of exclusive breastfeeding at 6 months can drop to lower than 20% in HICs, with a critically low 1% reported in an Australian national survey [33,36,37]. In comparison, low/middle-income countries (LMICs) reported slightly higher rates of exclusive breastfeeding at 6 months (36%) [33,37]. In conflict-affected countries, the median prevalence of exclusive breastfeeding has been reported to be 25% across 56 studies [37]. However, one study found that although breastfeeding rates are low when refugee-background women initially settle in HICs, those rates increase for each additional year living in the host country [38].

A variety of maternal sociodemographic factors, including age, socioeconomic status, education, and employment, have been shown to exert a notable influence on the likelihood of breastfeeding [39–41]. However, the individual effects and levels of significance of these determinants vary across HICs [42–45], suggesting the presence of country-specific confounders intertwined with cultural nuances, economic circumstances, and social infrastructure. Since SES is linked to employment status and educational attainment to the disadvantages of those with lower SES [46,47], future social interventions should have a special focus on women from low SES backgrounds. There is also a need to consider culturally specific programs for refugee-background populations, as they are often subject to discriminatory practices in both education and the workplace [15]. The literature provides mixed findings about the association between IPV and breastfeeding. IPV during pregnancy may affect breastfeeding directly, through physical stress hindering breastmilk release, or indirectly, via psychological barriers such as self-doubt, body negativity, male partner coercion, and depression [48]. While studies from Spain and the United States found an association between IPV and lower rates of breastfeeding [38,49,50], studies from Australia and Sweden reported no association [51,52]. On the other hand, almost all studies from LMICs, particularly in Asia and Africa, report associations between IPV and reduced rates of breastfeeding [26,53,54], as well as shortened duration of breastfeeding [55].

*Aim*

To inform future analyses, this study aimed to present preliminary findings on the effects of intimate partner violence and sociodemographic factors on ANE attendance and early breastfeeding in two cohorts: refugee-background women resettled in Australia and Australian-born women. These findings will provide valuable insights for policymakers and healthcare providers in developing comprehensive strategies to enhance the health and well-being of refugee mothers and children. The study proposed the following hypotheses: (1) refugee-background women would have lower ANE attendance rates and higher prevalence of IPV; and (2) individuals who were exposed to IPV and did not attend any ANE classes would have lower rates of breastfeeding on discharge from hospital. These hypotheses were formulated to guide the analysis and interpretation of the study's findings.

## 2. Results

### 2.1. Participants

A total of 1111 eligible women were interviewed at both T1 and T2, including 528 (47.6%) women born in Australia and 583 (52.5%) women who migrated from conflict-affected countries, referred to as refugee-background women in this paper (Table 1). The mean age of Australian-born women in our study was 29.1 (SD, 5.4) years, and the mean age of refugee-background women was 29.8 (SD, 5.4) years (Table 1). In total, 57% of the Australian-born women had completed a university degree or other post-school qualification, compared to 50.8% of women from refugee backgrounds. It can be assumed that some women from the latter group may have gained their qualifications from their country of origin. While the levels of education were similar between the two groups, there was a substantial disparity in the rates of employment. At T1, 60.0% of the Australian-born women were employed, while the employment rate was only 26.8% for refugee-background women. Among the Australian-born women, 34.2% were categorized as nulliparous, and this rate was 30.2% for refugee-background women.

**Table 1.** Sociodemographic characteristics, intimate partner violence, and antenatal education visits for women born in Australia and women born in conflict-affected countries.

| Variable | Australian-Born (*n* = 528) | | Refugee-Background (*n* = 583) | |
|---|---|---|---|---|
| | *N* | % | *N* | % |
| **Age groups [T1]** | | | | |
| <25 years old | 115 | 21.8 | 102 | 17.5 |
| 25–34 years old | 321 | 60.8 | 362 | 62.1 |
| ≥35 years old | 92 | 17.4 | 119 | 20.4 |
| Mean Age (SD) | | 29.1 (5.4) | | 29.8 (5.4) |
| **Education [T1]** | | | | |
| No post-school qualification | 225 | 42.6 | 287 | 49.2 |
| Diploma and vocational education | 136 | 25.8 | 99 | 17.0 |
| University degree | 167 | 31.6 | 197 | 33.8 |
| **Employment status [T1]** | | | | |
| Unemployed | 211 | 40.0 | 427 | 73.2 |
| Employed | 317 | 60.0 | 156 | 26.8 |
| **Intimate partner violence [T1]** | | | | |
| No | 391 | 74.1 | 330 | 56.6 |
| Yes | 137 | 25.9 | 253 | 43.4 |
| *N* | 528 | 100 | 583 | 100 |
| **Intimate partner violence [T2]** | | | | |
| No | 385 | 72.9 | 327 | 56.1 |
| Yes | 143 | 27.1 | 256 | 43.9 |
| *N* | 528 | 100 | 583 | 100 |
| **Parity [T1]** | | | | |
| Multiparous | 347 | 65.8 | 407 | 69.8 |
| Nulliparous | 180 | 34.2 | 176 | 30.2 |
| *N* | 527 | 100 | 583 | 100 |
| **Antenatal education [T2]** | | | | |
| No | 400 | 75.9 | 464 | 79.6 |
| Yes | 127 | 24.1 | 119 | 20.4 |
| Total | 527 | 100 | 583 | 100 |
| **Number of antenatal education classes [T2]** | | | | |
| 1–2 times | 94 | 75.2 | 52 | 46.4 |
| 1–4 times | 11 | 8.8 | 17 | 13.2 |
| 1–6 times | 15 | 12.0 | 21 | 18.8 |
| >6 times | 3 | 2.4 | 22 | 19.6 |
| Not stated | 2 | 1.6 | 0 | 0 |
| *N* | 125 | 100 | 112 | 100 |
| **# Designated classes [T2]** | | | | |
| Arabic and Sudanese pregnancy care classes | NA | | 47 | 56.0 |
| Multicultural antenatal classes | | | 24 | 28.6 |
| Others | | | 11 | 13.0 |
| Not stated | | | 2 | 2.4 |
| *N* | | | 84 | 100 |
| **Breastfeeding on discharge [T2]** | | | | |
| No | 96 | 18.3 | 62 | 10.7 |
| Yes | 430 | 81.7 | 520 | 89.3 |
| *N* | 526 | 100 | 582 | 100 |
| **Duration of breastfeeding [T2]** | | | | |
| <1 months | 187 | 36.9 | 195 | 34.9 |
| ≥1 months | 320 | 63.1 | 364 | 65.1 |
| Total | 507 | 100 | 559 | 100 |

SD: Standard deviation. NA: Not applicable to Australian-born women. *N*: Total number of interview answers.
# Designated classes: This question was directed to women born in conflict-affected countries only.

Refugee-background women reported higher rates of any IPV compared to Australian-born women at both time points. At T1, 25.9% of Australian-born women reported experiences of IPV in the past 12 months, whilst amongst women of refugee backgrounds, the rate of IPV experience was 43.4% (Table 1). Moreover, these rates increased slightly from T1 to T2 in both groups, with an additional 1.2% of Australian-born women reporting IPV at T2 (27.1%) and an additional 0.5% of refugee-background women reporting IPV (43.9%) at T2.

Australian-born women had higher ANE utilization compared to refugee-background women when measured by whether they had attended any ANE class (24.1% vs. 20.4%) (Table 1). Of the 119 refugee-background women who reported having attended ANE, up to 70.6% (84 out of 119) visited designated culture-specific or multicultural ANE classes.

Women from refugee backgrounds had higher breastfeeding rates compared to Australian-born women (89.3% vs. 81.7%, respectively). Both groups reported similar breastfeeding patterns: 34.9% of refugee-background women did not breastfeed or did for less than 1 month vs. 36.9% of Australian-born women; 65.1% of refugee-background women breastfed for more than 1 month vs. 63.1% of Australian-born women (Table 1).

## 2.2. Factors Associated with Antenatal Education Engagement

Results from bivariate analyses, presented in Table 2, show that the prevalence of ANE visits amongst Australian-born women was significantly higher for those who obtained post-school qualification ($p = 0.001$), were employed ($p = 0.001$), were nulliparous ($p = 0.001$), and reported no experiences of IPV during the perinatal period ($p = 0.006$). A similar trend between these factors and ANE attendance rates was also observed in refugee-background women. However, only higher educational attainment was statistically significant ($p = 0.003$).

**Table 2.** Association of sociodemographic characteristics and intimate partner violence with antenatal education (ANE) visits for women born in Australia and women born in conflict-affected countries.

| Sociodemographic Characteristics, IPV and Parity | Australian-Born (*n* = 528) | | | Refugee-Background (*n* = 583) | | |
|---|---|---|---|---|---|---|
| | *N* | ANE | | *N* | ANE | |
| | | *n* | % | | *n* | % |
| All | 527 | 127 | 24.1 | 583 | 119 | 20.4 |
| Age [T1] | | | | | | |
| <25 years old | 114 | 27 | 23.7 | 102 | 24 | 23.5 |
| 25–34 years old | 321 | 78 | 24.3 | 362 | 78 | 21.5 |
| ≥35 years old | 92 | 22 | 23.9 | 119 | 17 | 14.3 |
| *p* | | | 0.990 | | | 0.162 |
| Education [T1] | | | | | | |
| No post-school qualification | 225 | 33 | 14.7 | 287 | 42 | 14.6 |
| Diploma and vocational education | 135 | 41 | 30.4 | 99 | 24 | 24.2 |
| University degree | 167 | 53 | 31.7 | 197 | 53 | 26.9 |
| *p* | | | 0.001 | | | 0.003 |
| Employment status [T1] | | | | | | |
| Unemployed | 211 | 28 | 13.3 | 427 | 83 | 19.4 |
| Employed | 316 | 99 | 31.3 | 156 | 36 | 23.1 |
| *p* | | | 0.001 | | | 0.335 |
| Intimate partner violence [T1] | | | | | | |
| No | 391 | 106 | 27.1 | 330 | 75 | 22.7 |
| Yes | 136 | 21 | 15.4 | 253 | 44 | 17.4 |
| *p* | | | 0.006 | | | 0.113 |
| Parity [T1] | | | | | | |
| Multiparous | 347 | 27 | 7.8 | 407 | 42 | 10.3 |
| Nulliparous | 179 | 99 | 55.3 | 176 | 77 | 43.8 |
| *p* | | | 0.001 | | | 0.001 |

Adjusting for age, employment status, and other sociodemographic factors, adjusted odd ratios (AORs) from multiple logistic analyses, presented in Table 3, further predicted that for both groups, women with a post-school level of education were twice as likely to attend ANE classes (University degree—Australian-born, AOR: 2.69, 95% CI, 1.45–5.00, *p* < 0.01. Refugee-background, AOR: 2.61, 95% CI, 1.43–4.74, *p* < 0.01). Furthermore, nulliparous women in both groups were more likely to attend ANE classes (Australian-born, AOR: 14.57, 95% CI, 8.67–24.55, *p* < 0.01. Refugee-background, AOR: 6.45, 95% CI, 4.13–10.09, *p* < 0.01). Employment status and IPV exposure were not found to be significantly associated with ANE visits for any group of women (Table 3).

**Table 3.** Associations of sociodemographic characteristics and intimate partner Violence (IPV) with Antenatal Education attendance: Adjusted odds ratios (AORs) with 95% confidence interval (95% CI) from logistic regression analysis for women born in Australia and women born in conflict-affected countries.

| | Outcome Variables | |
| --- | --- | --- |
| | **Antenatal Education (no = 0, yes = 1)** | |
| **Significant Factors #** | **Australian-Born** | **Refugee-Background** |
| | **AOR (95% CI)** | |
| Education [T1] | | |
| No post-school qualification (RC) | 1.00 | 1.00 |
| Diploma and vocational education | 2.01 (1.08–3.73) * | 1.38 (0.75–2.56) |
| University degree | 2.69 (1.45–5.00) ** | 1.81 (1.08–3.04) * |
| Employment status [T1] | | |
| Unemployed (RC) | 1.00 | 1.00 |
| Employed | 1.06 (0.58–1.93) ** | 0.76 (0.45–1.28) |
| Any IPV [baseline] | | |
| No IPV (RC) | 1.00 | 1.00 |
| Any IPV | 0.54 (0.29–1.02) | 0.91 (0.57–1.45) |
| Parity | | |
| Multiparous (RC) | 1.00 | 1.00 |
| Nulliparous | 14.57 (8.67–24.55) ** | 6.45 (4.13–10.09) ** |

# Factors included in multiple logistic regression model were found to be statistically significant (*p* < 0.05) in bivariate analysis. RC, reference category; AOR, adjusted odds ratio; 95% CI, 95% confidence interval; * *p* < 0.05; ** *p* < 0.01.

### 2.3. Factors Associated with Breastfeeding Status

Results presented in Table 4 indicate that for both Australian-born and refugee-background women, the prevalence of breastfeeding at discharge was significantly higher for those who obtained post-school qualifications and were employed (*p* < 0.05). For both groups of women, the association of breastfeeding status with age and IPV exposure was not found to be statistically significant (*p* > 0.05). A positive association between ANE utilization and breastfeeding rates was observed in both groups of women. However, the data showed that this association was not statistically significant (*p* > 0.05).

AORs from multiple logistic analyses presented in Table 5 revealed that, amongst refugee-background women, none of the predictors were statistically significant (*p* > 0.05). However, Australian-born women with post-school level of education are three times more likely to be breastfeeding at discharge (diploma and vocational education—AOR: 3.26, 95% CI, 1.72–6.18, *p* < 0.01. University degree—AOR: 2.61, 95% CI, 1.43–4.74, *p* < 0.01).

**Table 4.** Association of sociodemographic characteristics, intimate partner violence, and antenatal education visits with breastfeeding rates for women born in Australia and women born in conflict-affected countries.

| Variable | Australian-Born (*n* = 528) | | | Refugee-Background (*n* = 583) | | |
|---|---|---|---|---|---|---|
| | *N* | Breastfeeding | | *N* | Breastfeeding | |
| | | *n* | % | | *n* | % |
| All | 526 | 430 | 81.7 | 582 | 520 | 89.3 |
| **Age [T1]** | | | | | | |
| <25 years old | 113 | 83 | 82.3 | 102 | 89 | 87.3 |
| 25–34 years old | 321 | 263 | 81.9 | 362 | 327 | 90.3 |
| ≥35 years old | 92 | 74 | 80.4 | 118 | 104 | 88.1 |
| *p* | | | 0.934 | | | 0.601 |
| **Education [T1]** | | | | | | |
| No post-school qualification | 224 | 161 | 72.3 | 286 | 247 | 86.4 |
| Diploma and vocational education | 135 | 121 | 89.6 | 99 | 89 | 89.9 |
| University degree | 167 | 147 | 88.0 | 197 | 184 | 93.4 |
| *p* | | | 0.001 | | | 0.047 |
| **Employment status [T1]** | | | | | | |
| Unemployed | 210 | 161 | 76.7 | 426 | 373 | 87.6 |
| Employed | 316 | 269 | 85.1 | 156 | 147 | 94.2 |
| *p* | | | 0.014 | | | 0.021 |
| **Intimate partner violence [T1]** | | | | | | |
| No | 391 | 318 | 81.3 | 330 | 297 | 90.0 |
| Yes | 135 | 112 | 83.0 | 252 | 223 | 88.5 |
| *p* | | | 0.672 | | | 0.559 |
| **Parity [T1]** | | | | | | |
| Multiparous | 346 | 276 | 79.8 | 406 | 360 | 88.7 |
| Nulliparous | 179 | 153 | 85.8 | 176 | 160 | 90.9 |
| *p* | | | 0.109 | | | 0.421 |
| **Antenatal Education [T2]** | | | | | | |
| No | 399 | 321 | 80.5 | 463 | 408 | 88.1 |
| Yes | 127 | 109 | 85.8 | 119 | 112 | 94.1 |
| *p* | | | 0.172 | | | 0.059 |

**Table 5.** Associations of sociodemographic characteristics and intimate partner violence (IPV) with breastfeeding at discharge: adjusted odds ratios (AOR) with 95% confidence interval (95% CI) from logistic regression analysis for women born in Australia and women born in conflict-affected countries.

| | Outcome Variables | |
|---|---|---|
| | Breastfeeding (no = 0, yes = 1) | |
| **Significant Factors #** | Australian-Born | Refugee-Background |
| | AOR (95% CI) | |
| **Education [T1]** | | |
| No post-school qualification (RC) | 1.00 | 1.00 |
| Diploma and vocational education | 3.26 (1.72–6.18) ** | 1.27 (0.60–2.68) |
| University degree | 2.61 (1.43–4.74) ** | 1.86 (0.93–3.71) |
| **Employment status [T1]** | | |
| Unemployed (RC) | 1.00 | 1.00 |
| Employed | 1.32 (0.80–2.19) | 1.89 (0.87–4.09) |

# Factors included in multiple logistic regression model were found to be statistically significant (*p* < 0.05) in bivariate analysis. RC, reference category; AOR, adjusted odds ratio; 95% CI, 95% confidence interval; ** *p* < 0.01.

### 3. Materials and Method

Ethics approval: The longitudinal WATCH cohort study was approved by the South Western Sydney Local Health District Human Research Ethics Committee (HC13049) and Monash Health Ethics Committee. Participants gave written informed consent and were remunerated for their time. The study included extensive training of research staff derived from the same cultural and language backgrounds as the target populations, followed by tests of interview competence (rater reliability). Staff received training for IPV, sensitive interview techniques, research methods, and the use of World Health Organization diagnostic measures for IPV. The study also followed the WHO protocol for ensuring the safety of participants who may have experienced IPV and applied a recognized approach for designing and testing measures that are not in English.

#### 3.1. Study Design

This study analyzed the baseline (T1) and first-follow-up (T2) data of 1335 women who participated in the Women Aware with Their Children (WATCH) study. The baseline survey was undertaken between January 2015 and March 2016, and follow-up occurred approximately six months after the birth of the child. The study design and methods are fully described in previous papers [27,56].

The primary study was undertaken at three large public hospital antenatal clinics, two in Sydney and one in Melbourne. Women from refugee-background were systematically invited to participate in the study as part of the refugee cohort if they were identified to be from Arabic-speaking, Sudanese, or Sri Lankan Tamil backgrounds. These three groupings ensured a good representation of the global refugee intake entering Australia at the time of data collection. The criteria for participation were not limited to the type of visa held. Women born in Australian-born were recruited at the same time and from the same hospitals using a randomized selection process. Data for the current analysis are from two time points (T1 is the first trimester of pregnancy, and T2 is 6 months post-partum). Finally, the present study consisted of 1111 women who participated in the primary interviews.

#### 3.2. Data Collection and Measures

All data for this secondary study, including the demographics, were obtained from the WATCH study database for the specific and planned research analysis of IPV, ANE attendance, and breastfeeding.

Recruitment and the baseline interview (T1) occurred at, or close to, the participant's first appointment at the antenatal clinic (most occurred between 12 and 20 weeks of gestation). Follow-up interviews (T2) were conducted at home, either in person or by telephone, approximately 6 months after the birth of the index child. At baseline (T1), the response rate was 84.8% (1335 women out of 1574); at T2, the retention rate was 83.2% (1111 out of 1335 interviewed at T1) [56]. The analytical sample of this secondary study included 1111 out of 1335 women who participated in the interviews.

Measures related to IPV were included at both T1 and T2 interviews. At T2, standardized Local Health District measures related to pregnancy and childbirth were added: antenatal care uptake, antenatal clinic visits, and breastfeeding status.

Measures were subjected to rigorous assessment of cultural and linguistic accuracy in the languages used, including standard translation, back-translation, and assessment and refinements by groups of linguistic and cultural experts [27,56].

#### 3.2.1. Sociodemographic Characteristics

Items recording age, marital status, level of education and qualification, household composition, employment, and housing status were consistent with the Australian National Census. These items can be benchmarked against the Australian population. Countries of birth for inclusion in the study (all Arabic-speaking countries, Sudan, and Sri Lanka) were identified by clinic records, requests for an interpreter, or culturally recognizable surnames, and country of birth was checked again at the time of recruitment. Many people arrive

from conflict-affected settings on visas other than special humanitarian visas, which are therefore not accurate reflections of being a refugee. For this analysis, we have included all recruited women who were born in conflict-affected countries, whom, in this paper, we refer to as refugee-background women.

### 3.2.2. Parity

Parity was assessed during the baseline survey. For this study, women having no previous births reported at baseline were categorized as 'nulliparous women', and women having had at least one previous birth were categorized as 'multiparous women'.

### 3.2.3. Intimate Partner Violence

IPV was assessed using items from the WHO Violence Against Women questionnaire, which asks about physical, psychological, and sexual violence perpetrated by the current or most recent partner in the past 12 months [56]. For this study, women were assigned to two IPV categories: (1) No IPV; (2) Any IPV (either psychological and/or physical IPV; psychological IPV includes jealousy or anger if she talks to other men, accusations of being unfaithful, not permitting meetings with female friends, limiting contact with family, insisting on knowing the woman's whereabouts, humiliating her in front of others, threatening harm to her or someone close to her; physical abuse includes pushing, shaking, throwing items, slapping, twisting arm, punching, kicking, dragging, strangling, burning, threats with a knife, gun, or other weapon, and attacks with a knife, gun, or other weapon).

### 3.2.4. Antenatal Education Attendance

Survey answers were collected regarding whether the participant attended any ANE sessions (yes/no), as well as the number of antenatal classes attended, and whether they attended ANE classes specifically offered for women from mainly non-English-speaking backgrounds, including ANE offered at the Blacktown Hospital site for women from Sudanese and Arabic-speaking backgrounds.

### 3.2.5. Breastfeeding

Women were asked at T2, "Were you breastfeeding on discharge? After discharge, how long did you breastfeed up to this point?" These are standard questions asked by NSW Health (Australia's largest public health system) on discharge after the birth of the child.

### 3.3. Statistical Analysis

Descriptive statistics of participants' characteristics (age, educational attainment, employment status, parity, prevalence of IPV, ANE visits, and breastfeeding status) of those who attended both T1 and T2 surveys were explored for both groups of women. Bivariate (cross-tabular) and multiple logistic regression analyses were performed to examine the association of sociodemographic factors, parity, and IPV exposure with ANE visits. Potential risk factors for ANE visits found to be statistically significant ($p < 0.05$) in bivariate analysis were included in multiple logistic regression analyses. The aim of multiple logistic regression analysis was to estimate the relative contributions of each significant risk factor to ANE visits. Further, we also performed bivariate and multiple logistic regression analysis to explore the association of sociodemographic factors, IPV exposure, and ANE visits with breastfeeding status at discharge. Results of bivariate analyses are presented as percentages and means; chi-square ($\chi2$ ) was applied to examine the significant differences across sub-groups. The adjusted odds ratios (AORs) from logistic regression analysis with their 95% confidence intervals (95% CI) are shown to express the relative contributions of each potential risk factor to likelihood of ANE visits and breastfeeding status, adjusted for the effects of other variables in the model. All the analyses were carried out separately for both Australian-born and refugee-background women. The analyses were conducted with SPS S v. 27 [57].

## 4. Discussion

Despite the known benefits of maternal awareness and agency associated with ANE, there remains a critical gap in evidence regarding the prevalence of ANE uptake and its impact on maternal and child outcomes, particularly amongst women from refugee backgrounds. This population may encounter unique challenges, including increased susceptibility to complications related to prior trauma, IPV, and psychosocial adversity [56,58]. Our study is large and methodically rigorous, enabling us to compare the prevalence of and associations between ANE attendance, IPV exposure, early breastfeeding rates, and various other sociodemographic factors. The uniqueness of our study lies in its focus on women from refugee backgrounds who resettled in Australia, allowing us to make meaningful comparisons regarding their experiences in antenatal care to those of women born in Australian. The study follows a cohort design, and data for this analysis were drawn from two relevant time points (the first trimester of pregnancy and the post-partum period). The findings provide important insights for antenatal clinicians and policymakers.

### 4.1. Antenatal Education

ANE attendance was similar for both groups when measured categorically by any attendance. The rates for the utilization of ANE amongst Australian-born women were significantly lower (24.1%) compared to findings from past studies in both Australia (89%) [59] and the U.K. (53.1%) [60]. Rates of ANE attendance can vary by country of birth [11] and psychosocial factors [61], suggesting that the lower rates in our study may be attributed to the recruitment of women living in lower-socioeconomic-status areas of Sydney and Melbourne [61]. Women from refugee backgrounds attending ANE participated in a higher number of antenatal classes: 19.6% of women from refugee backgrounds visited ANE classes more than 6 times, with an average of 1–4 visits. Directly comparing the number of classes attended by Australian-born and refugee-background women was challenging due to the disparity in the number of classes offered. Australian-born women attending the standard programs had access to approximately six classes in our study. In contrast, women from mainly non-English-speaking backgrounds, including refugee-background women, were offered up to 21 classes. The notable number of ANE classes attended by women from refugee backgrounds is, regardless of comparison, indicative of a positive experience. We posit that the high number of attendances per person reflects the culturally sensitive and supportive nature of the specialized ANE programs run for mainly non-English-speaking-background women (attended by 70.6% of the sample). Although further research is required to fully explore this observation, our finding is a novel and noteworthy finding regarding the pivotal role of culturally and linguistically specific ANEs in enhancing healthcare accessibility. Notably, the Arabic and Sudanese Pregnancy Care Clinic at Blacktown Hospital in Sydney, which was one of our recruitment sites and is the site for a current qualitative study [27], is a prominent example (attended by 58% of the sample). The emergence of such specialized clinics catering to the unique needs of diverse populations holds the potential to close healthcare gaps and promote culturally responsive services.

### 4.2. Social Determinants

Antenatal services are predominantly attended by women with higher levels of education and from the middle-to-upper socioeconomic strata across various developed nations, namely Canada and the United States [62,63], South Korea [64], and Belgium [65]. Regardless of immigration background, women attaining higher levels of education typically exhibit greater health literacy and autonomy in navigating their pregnancy [66,67]. As such, our study confirms that maternal educational level is the universal and most predictive determinant of ANE utilization, encompassing both women from refugee backgrounds and native Australians. Higher parity has a negative effect on ANE attendance, and this resonates with previously published literature on adequate antenatal care attendance [68–70]. First-time mothers may be encouraged or motivated to learn to care for themselves and their unborn child, whereas parous women may not perceive the ANE as a necessity, given that they are

well "experienced" with previous pregnancies, especially if they were uncomplicated [71]. The higher uptake of ANE amongst employed women in the Australian-born cohort, as observed in our study, aligns with similar findings in a recent study in Belgium [65] and in older literature across HICs [72,73]. However, a recent study in the United Arab Emirates did not find any direct associations between employment status and antenatal visits [74], suggesting that contemporary work-related issues may engender barriers to attending clinic appointments [75]. These issues may encompass limited work time flexibility, including insufficient time off for medical visits, and greater job demands hindering a women's ability to prioritize their health [76]. Women occupying higher professional and executive roles are more likely to face these challenges [76], a factor for which our study did not examine. The recruitment of women from a lower-SES region in Sydney and Melbourne, therefore, may have resulted in a sample of women having fewer occupational demands and better attendance to health needs [67]. While the association between employment and ANE attendance amongst refugee-background women is weak and statistically insignificant in our study, it suggests a potential association worth further exploration. Nonetheless, there is limited contemporary research on the specific impact of working during pregnancy and consideration of workplace culture in HICs, including the magnitude of workplace modification to cater to the unique needs of pregnant women. Our study highlights the potential benefits of employment during pregnancy and emphasizes the need to further explore work-related factors that can facilitate healthcare-seeking behavior.

Health services and support provided in HICs are typically less accessible or culturally sensitive, particularly for women facing social and economic marginalization, including those from refugee backgrounds [15,77]. Our study reinforces the significance of sociodemographic factors in predicting ANE attendance and underscores the need to address barriers to healthcare access that are influenced by economic disadvantage, lower educational levels, and visa status. This exploration will be instrumental for the design of targeted health interventions for women from culturally diverse backgrounds.

### 4.3. Intimate Partner Violence

We report a high number of pregnant Australian women, refugee-background and Australian-born, have experienced IPV. Data in Roman-Galvez and colleagues' systematic review [78] showed that the highest range of any kind of IPV during pregnancy (including sexual, physical and emotional) was reported in Australia (15.4–40%), along with Portugal and the USA. Our study confirms this broad range in findings, suggesting that specific subpopulations in the same region can be at increased risk of experiencing IPV during pregnancy [21]. General rates are lower when compared to another study focusing on any IPV in women of refugee backgrounds (79.8%) [79], highlighting the urgent need to address the alarming risk faced by this population, which can engender severe and lethal consequences. IPV during pregnancy is associated with serious negative outcomes for maternal and child health [80,81]. The most described adverse physical health impacts associated with IPV in the literature include maternal death, pregnancy complications, and stillbirth [16,21]. While our protocol measured IPV experience within the last 12 months, we were unable to assess whether women in our study were exposed to IPV during their pregnancy. Nevertheless, IPV-related trauma can directly impair a women's functioning before, during, and after birth. The risk is particularly high for women from conflict-affected countries who face unique risk factors, including trauma before arrival in the settlement country [27], lack of social support, and increased dependency on their intimate partners after the resettlement [82]. Further, our study shows a slight increase in IPV rates from the women's first trimester to six months post-partum in both groups. This concerning finding postulates either new perpetrations during or after the pregnancy or underreporting of IPV at T1, and it emphasizes the need to strengthen IPV screening tools and intervention programs during antenatal care. To prevent detrimental harm to the women and their babies in the perinatal period, there is a dire need for awareness

and interventions for IPV amongst pregnant women, with a focus on women arriving in Australia from conflict-affected settings.

This study is unique in that it explores the relationship between perinatal IPV and the utilization of antenatal education. To the best of our knowledge, this is the only study that investigates this relationship, comparing the correlations in two distinctive cohorts of Australian-born and women from conflict-affected countries. The findings from our study suggest that women who have experienced any form of IPV by a former or current partner, whether it occurred before or during pregnancy, were significantly more likely to receive inadequate ANE by way of lower attendance compared to women who reported no IPV. This association was observed regardless of the women's background.

Although a causal link is unable to be established from this preliminary analysis, IPV may prevent women from accessing ANE, either because of a coercive and controlling partner hindering a woman's attendance, psychological distress and impaired functioning, or financial hardship. For example, previous studies have reported IPV to be significantly associated with depression, anxiety, and suicidal ideation and related poor functioning [18,83,84]. Further studies show that IPV reduces decision-making power and creates financial barriers [85]. Despite underreporting, minority or migrant women, including refugee-background women, experience higher rates of IPV during pregnancy [86], a factor that is consistent with our findings of IPV prevalence. Women from refugee backgrounds may have lower socioeconomic status, fewer social supports, and higher rates of mental disorders, including depression. Moreover, refugee-background women may also experience specific factors that may further lower the likelihood of attending ANE: for example, lack of trust in authorities, trauma related to war and conflict, and poor English language skills [27]. This supposition supports the finding that women experiencing IPV are less likely to attend ANE. Of great interest is that women from refugee backgrounds who did attend ANE, regardless of IPV status, attended several classes, indicating that they enjoyed or benefited from the experience. We also note the significance of 70.6% of our refugee background cohort having attended an ANE designed for women from mainly non-English-speaking backgrounds, a service that may resonate with refugee-background women because of the qualitatively described appreciation of cultural and linguistical familiarity provided by such ANE programs at the hospitals from which the participants were recruited. This is a current area of inquiry for our team.

Our findings confirm the importance of antenatal services such as ANE as sites for IPV identification, prevention, and intervention, as well as the need for specialized assessments and ANE programs for women from refugee backgrounds.

*4.4. Breastfeeding*

We report a positive correlation between ANE attendance during pregnancy and early breastfeeding on discharge. Although the association was not statistically significant in either cohort, the positive health correlates of breastfeeding for women and their babies reinforce the inherent value of reporting an association. These findings are also consistent with previous studies that have examined the effect of utilizing antenatal services and breastfeeding education on the rates of early breastfeeding initiation and continuation [41,87–93]. Knowledge about breastfeeding gained through maternal health services, such as ANE, may help mothers to overcome concerns related to breastfeeding [94,95] and encourage them to favor breastfeeding over other types of infant feeding [96].

We found that despite higher IPV prevalence and lower ANE attendance, women from refugee backgrounds were slightly more likely to initiate breastfeeding soon after birth. However, it should be noted that the rates of breastfeeding initiation in both cohorts still remain lower when compared to a national survey conducted in Australia, which reported a prevalence of 98% for breastfeeding initiation, of which 93% of infants were exclusively breastfed [36]. Cultural views and norms related to breastfeeding are important to understand as factors that may impact breastfeeding, in addition to any information provided during ANE. A study found that mothers may make the decision about breastfeeding long

before conception, based on cultural beliefs [97]. For example, breastfeeding for 2 years is recommended in the holy book (Qur'an) in Islam, and therefore, the desire to breastfeed amongst Middle Eastern women is deeply rooted in their cultural values and the belief that they will receive support from the woman's partner and her community [98]. The evidence shows that breastfeeding is more widely practiced in LMICs than it is in most HICs [33] and that women from LMICs who migrated to HICs may not change their breastfeeding patterns [99]. Not all non-Western cultures, however, continue to breastfeed at higher rates after migration. It should be acknowledged that we did not measure either group's adherence to the recommended duration of exclusive breastfeeding for 6 months' duration. However, it is worth noting that the rates of discontinuing breastfeeding within the first month were significantly high, reaching 34.9–36.9%. An Australian study showed that Vietnamese refugee-background women had higher ANE attendance rates but lower rates of breastfeeding compared to Australian-born women, mostly due to cultural traditions [100]. Another study in California also reported a lack of interest in obtaining information on breastfeeding amongst Southeast Asian women from refugee backgrounds attending ANE [101]. Our study highlights the importance of understanding cultural differences and the need for ANE content to be adapted for the specific population. We recommend that ANE is delivered by bicultural or bilingual workers from relevant backgrounds to ensure diverse cultural practices and norms of the target demographics are reflected.

This study goes some way to addressing the paucity of evidence on ANE and its association with socioeconomic factors, IPV, and breastfeeding practices. With increasing numbers of economic and humanitarian migrants entering Australia, our study suggests the need for ANE programs that are specific to culturally diverse groups. There is a critical need to adopt trauma-informed approaches when caring for expecting mothers, taking into consideration the impact of IPV and conflict-related trauma, during both pregnancy and the post-partum period. All women should be screened for IPV in the antenatal setting, and those who disclose IPV should be provided with additional support to access ANE classes. When appropriate, referral to culturally appropriate and accessible domestic violence services should also be provided. Furthermore, to address barriers to disclosure amongst women with difficulties reporting their partners, future ANE planning should include access to IPV wraparound services. Given the high prevalence of IPV amongst pregnant women attending ANE in our study, all healthcare providers in the ANE setting should receive training consistent with a trauma-informed approach. This will enable them to identify signs of IPV and respond appropriately.

*4.5. Strength and Limitations*

We performed a large, rigorous, systematically recruited study of women at two time points in the perinatal period. The study included measures for ANE, IPV, breastfeeding rates on discharge, and sociodemographic characteristics. One of the notable strengths of the study is the data for both a population with refugee backgrounds and one that is Australian-born, which is rare to find in the current literature. The IPV questions relate to the current or most recent relationship in the past 12 months, which means that we cannot assume the presence of current IPV at the time of the interview or during the pregnancy. Despite using two time points, the study is cross-sectional, and associations are indicative but cannot demonstrate causation. It should be acknowledged that some findings reported in our study did not reach statistical significance. However, despite these limitations, the findings provide valuable preliminary insights and associations that contribute to the existing knowledge on the topic of obstetric health among refugee women in high-income countries. The study prompts the need for future research to validate and confirm our findings.

**5. Conclusions**

Our findings confirm a higher prevalence of IPV and lower ANE uptake among women from refugee backgrounds compared with women born in Australia. These results warrant attention to ANE access and support for refugee-background women (who were

exposed to war-related conflict and have resettled in developed countries). In both groups of women, higher education and nulliparity are better predictors of ANE attendance than employment, although all are highly associated with increased rates of ANE class attendance. Women exposed to any kind of IPV (emotional, sexual, or physical IPV) tend to have lower ANE attendance rates. Although it is not statistically significant, poor ANE utilization reduced breastfeeding rates. The novel and summary finding is that being from refugee backgrounds, a single report of IPV, lower educational attainment, and unemployment define subpopulations of women at higher risk for lower utilization of ANE and/or lower likelihood of breastfeeding. This indicates that checking sociodemographic and psychosocial information at the antenatal clinic and subsequent screening and support for higher-risk women may help avert negative pregnancy and childbirth outcomes.

**Author Contributions:** Conceptualization, S.R., J.F., N.N.; Investigation, N.N., B.M., F.H., B.K., M.Y., Y.K.; Formal analysis, T.A.N., M.M., S.R.; Writing—original draft, T.A.N., S.R.; Writing—review and editing, T.A.N., S.R., M.M., J.F., M.K., N.N., B.M., F.H., B.K., M.Y., Y.K.; Supervision, S.R., M.K.; Funding acquisition, S.R., J.F. All authors have read and agreed to the published version of the manuscript.

**Funding:** This research was funded by the National Health and Medical Research Council, Australia, grant numbers GNT1086774 and GNT1164736.

**Institutional Review Board Statement:** The study was conducted in accordance with the Declaration of Helsinki and approved by the South Western Sydney Local Health District Human Research Ethics Committee (HC13049) and Monash Health Ethics Committee.

**Informed Consent Statement:** Written informed consent has been obtained from the patient(s) to publish this paper.

**Data Availability Statement:** Data are available upon reasonable request from the corresponding author.

**Acknowledgments:** We acknowledge the women participants in the WATCH study, Anggy Duarte for managing data entry, and Gordana Sobacic at Southwest Sydney Local Health District for administrative support.

**Conflicts of Interest:** The authors declare no conflict of interest.

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
