# Peer review of "Determinants of Antenatal Education and Breastfeeding Uptake in Refugee-Background and Australian-Born Women"

_women, doi:10.3390/women3020020_

Round 1

Reviewer 1 Report

Dear Authors, this paper about the prevalence and association between antenatal education attendance, breastfeeding and sociodemographic factors  among women born in Australia  and from refugee backgrounds is really interesting and well written. I am sure it will help both scientists and dental workers to improve their knowledge .

Nevertheless some issues need to be solved before its final acceptance in the journal.

Title: please simplify the title, it is too long

Abstract: divide this part into: introduction, materials and methods, results, conclusions

Introduction: This part is really important, it helps the reader to deep into the research subject. I would suggest to not divide the introduction into chapters, this is misleading for the reader. Moreover you should add some informations about breast-feeding difficulties and frenotmy. This paper can help you: Colombari GC, Mariusso MR, Ercolin LT, Mazzoleni S, Stellini E, Ludovichetti FS. Relationship between Breastfeeding Difficulties, Ankyloglossia, and Frenotomy: A Literature Review. J Contemp Dent Pract. 2021 Apr 1;22(4):452-461.

Materials and methods: The ethical approval should not be stated under the "chapter" statistical analysis. 

results and discussion are well written but badly divided. The common reader might have some difficulties in reading this paper. I strongly recommend to divide the paper, as the abstract, in: introduction, materials and methods, results, discussion, conclusions.

Author Response

Dear reviewer, 

Thank you for taking the time to review our research paper and providing valuable suggestions. We appreciate your feedback, and we have made several changes based on your recommendations. Please see attached for the edited version. 

Firstly, we have revised the title of the manuscript to "Determinants of Antenatal Education and Breastfeeding Uptake in Refugee-background and Australian-born Women," as you suggested. We believe this new title better reflects the scope and focus of our study.

Regarding the abstract, we have taken your advice and divided it into distinct sections, including introduction, materials and methods, results, and conclusions. This organization will enhance the clarity and readability of the abstract, making it easier for readers to grasp the key aspects of our research.

In response to your comment about the introduction section, we have removed the subheadings as per your suggestion. By doing so, we have ensured a more cohesive flow of information in the introduction, allowing readers to follow the logical progression of our research.

Regarding the article on breastfeeding difficulties and frenotomy, we agree with your viewpoint that it is a very interesting and important piece of literature. However, it is tangential to our research question. Therefore, we have decided not to include it in our paper, maintaining a stronger focus on the sociocultural determinants of antenatal education and breastfeeding uptake.

Additionally, we have incorporated your suggestion of moving the ethics approval to the top of the Material and Methods section. 

With regard to the division of the results and discussion sections, we would like to clarify that we had already structured the paper as you suggested. Therefore, we have made no changes to the sequence of our subheadings in these sections.

Once again, we sincerely appreciate your thoughtful suggestions. We are confident that these revisions have strengthened the quality and clarity of our research paper. Thank you for your valuable input, which has undoubtedly contributed to the improvement of our manuscript.

Reviewer 2 Report

The authors have addressed a very important topic of public health, including  the role of antenatal education, breastfeeding, exposure to intimate partner violence and in general women's mental health, which fits well into the thematic scope of the journal and this special issue.
Overall, this is an interesting manuscript however, there are several areas that need to be addressed to help improve the paper prior before final publication.

Please find below the areas for improvement:
Point 1: The abstract should be improved, the introduction is too long and however there is not enough information about the research methodology and specific results for this research.

Point 2: Keywords: I suggest to add women or mother.

Point 3: Introduction: The data contained in the introduction are important due to the purpose of the study - however, due to its length, I suggest shortening it. Some of the content from this section can be moved to the discussion, which would significantly strengthen it.
Lines 96-98: this sentence is incomplete - it should be:
.......for six months with continued breastfeeding until the age of 2 years or later.
Subsection 1.5 - this is the research goal and hypotheses that can be combined.
The following hypotheses were put forward (1).....and (2)....

Point 4: Materials and Methods are generally well described, however, for better readability, I suggest adding the flowchart of study design and sample collection.
Lines 167-168:  I propose such a record: Finally, the present study consisted of 1,111women who participated in the interviews.
Line 169: This sentence is also unclear.
Subsections:  Ethics approval - it should be earlier.

Point 5: The chapter/section Results requires more consideration.
Pay attention to the order of the tables in question - it may be better to combine tables 1 and 2.
In addition, I did not find an explanation for the different number of N in tables 1, 2 and 3.

Point 6: In my view, the discussion section would benefit from the addition of more references to the literature and a deeper discussion in relation to the existing studies mentioned in the introduction and the consequences for the child and mother. The discussion requires refinement, highlighting the strengths of the study and giving what's novelty.

In addition, the manuscript should be adapted to the editorial guidelines of the journal, especially with regard to citing references. In my opinion, not all references are cited in the manuscript - Authors should check this carefully.

Author Response

Dear Reviewer,

Thank you for your detailed review and valuable suggestions regarding our research paper. We appreciate your time and effort in providing constructive feedback. We have carefully considered your comments and made the necessary revisions to improve the quality and clarity of the manuscript. Please see the attachment for the reference.

Regarding the abstract, we have addressed your concern about its length by shortening the introduction and reducing the abstract to 203 words. We have emphasized the use of multi-logistic regression in exploring the bivariate associations between our outcome measures, as well as the cross-sectional nature of our survey. Additionally, we have included the prevalence results for ANE attendance, breastfeeding, and IPV. Due to the word limit, we couldn't include the adjusted odds ratios in the abstract. However, we have reported the significant associations in a concise manner. We have also added  "women" as a keyword, as per your suggestion.

In response to your comment on the length of the introduction section, we have taken into account conflicting views from other reviewers and decided to prioritize providing more background knowledge on the topic. However, we have made appropriate cuts, and removed subheadings and replacing lines 156-160 of subsection 1.5 (now subheading titled "Aim") with a succinct outline of our aim.

Regarding the WHO recommendation on breastfeeding, we can confirm that our sentence is complete and aligned with the guidelines. We have reviewed the specific sentence you mentioned and confirmed its accuracy.

In the material and methods section, we have incorporated your recommended addition of "Finally, the present study consisted of 1,111 women who participated in the interviews" (lines 206-208). We have also clarified the previously unclear sentence (now line 220). Furthermore, we have moved the ethics approval to the top of the Material and Methods section for better organization.

Regarding your suggestion to include a flowchart for our study method, we acknowledge its potential value in enhancing the understanding of our research design and recruitment process. However, we have referenced our previous papers that provide detailed information on the study recruitment and full protocol. By citing these sources, we ensure that interested readers can access the comprehensive information they require.

Regarding the results section, we have made changes to the sequence and content of our tables. Table 1 and Table 2 remain the same, while what was previously Table 4 has been split into two distinctive tables presenting the adjusted odds ratios for ANE and breastfeeding. The sequence of the tables is as follows: Table (1) – participant characteristics and prevalence of outcome measures, (2) determinants of ANE, (3) adjusted odds ratios for determinants of ANE, (4) determinants of breastfeeding, and (5) adjusted odds ratios for determinants of breastfeeding. We believe this arrangement will alleviate any confusion for readers.

We have also provided clarification regarding the different numbers of N in Table 2. The numbers reflect the total answers received from women regarding their attendance at ANE. For example, in Table 2, there are n=528 Australian-born women, but we recorded N=527 survey answers on ANE attendance. Similarly, for refugee-background women, we had n=583 and N=583 answers. Since we investigated the associations between ANE and other factors, we only included data from women who answered questions about their ANE attendance in the analysis. The same rationale applies to Table 3 (now Table 4) for breastfeeding data.

In the discussion section, we have included comparisons with other studies, including those mentioned in our introduction. Please refer to lines 501-513 for the comparisons related to our IPV results and lines 589-592 for data on the prevalence of breastfeeding initiation. We have also provided further comments on the correlation between ANE attendance and parity (lines 453-458) and employment status (lines 461-469). Moreover, we have refined the discussion, highlighting the strengths of the study and emphasizing its novelty. Please see the paragraph under the subheading "Strengths and Limitations" in the Discussion section and the concluding paragraphs of each subheading for these improvements.

Once again, we sincerely appreciate your thoughtful suggestions. We are confident that these revisions have strengthened the quality and clarity of our research paper. Thank you for your valuable input, which has undoubtedly contributed to the improvement of our manuscript.

Best regards,

Reviewer 3 Report

Thank you for the opportunity to review the manuscript titled “

Prevalence and Associations between Antenatal Education attendance, Breastfeeding, and sociodemographic factors amongst women born in Australia and from refugee backgrounds”.

The authors conducted the research to check the factors affecting the frequency of using antenatal education (ANE), and breastfeeding among Australian and refugee women. In the study, the authors asked the participants about exposure to intimate partner violence (IPV), which additionally increases the value of the study. Noteworthy is the high ethics of the study, the more so that it concerns a very sensitive sphere.

The following are my comments describing these issues.

1.         The title

Maybe think about a better manuscript title, it's only a suggestion like

Antenatal Education, Breastfeeding, and Intimate Partner Violence among ..

2.         Introduction

In the introduction, authors should provide data on:

 How is the National health system in Australia organized, Antenatal Education in Australia, how many visits, and who pays the patient? e.t.c. How can refugee women learn about ANE?

Line 81. The authors submitted the article to an international journal, so they should write the data on how IPV looks in the world compared to Australia. Please add 1-2 sentences. Where is the highest and the lowest percentage of IPV?

 Line 89. Women from refugee backgrounds experience significantly higher rates of IPV than women born in the host country- how many percent?

 The introduction needs some corrections.

2. Results

Table 1. There are the results – Duration of breastfeeding < 1 month and ≥ 1 month

Line 96 The authors wrote: Early and long-term breastfeeding, including exclusive breastfeeding for six months, and non-exclusive continuation for two years has been recommended by WHO due to its numerous health benefits for both babies and mothers.

Therefore, none of the results below 1 and above 1 are correct.

So, what percentage of women in the study breastfed for more than 6 months to 2 years?

3. Discussion

Extensive references give you the opportunity to choose and improve your discussion.

In the discussion, the authors should more often compare the results of the presented study with other studies by other authors.

The discussion needs some improvement.

Author Response

Dear reviewer,

Thank you for taking the time to review our research paper and providing valuable suggestions. We appreciate your feedback, and we have made several changes based on your recommendations. Please see attachment for the edited version. 

We have now changed the title to "Determinants of Antenatal Education and Breastfeeding Uptake in Refugee-background and Australian-born Women." We believe this revised title better reflects the focus and scope of our research.

Regarding the introduction, we have included brief information on the national health system of Australia, specifically Medicare, and its accessibility for refugee women. You can find this information in the Introduction section, lines 53-57 (with the option to "Show all mark up" in the review toolbar in MS Word). Additionally, we have incorporated more literature on the rates of IPV across countries (lines 93-94) and the rates of IPV during pregnancy among refugee-background women (line 104) to enhance the context in our introduction.

In response to your comment about the percentage of women breastfeeding for more than 6 months to 2 years in the results, we acknowledge the limitation of our data. We were unable to record the exact duration of breastfeeding for each woman, as our data only categorized the duration as more or less than 1 month. Consequently, our main outcome measure focused on early breastfeeding initiation on discharge, rather than the prevalence of exclusive breastfeeding for 6 months. The data provided in Table 1 aims to offer a qualitative comparison of the breastfeeding patterns between our two cohorts. To address this potential confusion for readers, we will further discuss this matter in the discussion section. Please refer to lines 599-602 in the discussion section under the breastfeeding subheading.

In the discussion section, we have made efforts to draw comparisons between our results and other studies whenever possible. You can find these comparisons in lines 501-513 regarding IPV results and lines 589-592 regarding the prevalence of breastfeeding initiation in our cohorts compared to other reported data in Australia. We have also added further comments on the correlation between ANE attendance and parity (lines 453-458) and employment status (lines 461-469). Moreover, we have refined the discussion, emphasizing the strengths of the study and highlighting its novelty. Please see the subheading "Strengths and Limitations" in the Discussion section and the concluding paragraphs of each subheading for these improvements.

We genuinely appreciate your thorough review and insightful suggestions, which have undoubtedly enhanced the quality and clarity of our research paper. Thank you for your valuable input, and we believe these revisions have strengthened the overall manuscript.

Best regards,

Round 2

Reviewer 2 Report

Thank you for your effort regarding the amendments  that I previously pointed out. A great effort was made by the Authors to address all the questions.

This manuscript has been greatly improved and I can now support its application.

Kind regards